# Assessment of Wild Rocket (*Diplotaxis tenuifolia* (L.) DC.) Germplasm Accessions by NGS Identified SSR and SNP Markers

**DOI:** 10.3390/plants11243482

**Published:** 2022-12-12

**Authors:** João M. Reis, Ricardo J. Pereira, Paula S. Coelho, José M. Leitão

**Affiliations:** 1MED—Mediterranean Institute for Agriculture, Environment and Development, Faculdade de Ciências e Tecnologia, Campus de Gambelas, Universidade do Algarve, 8005-139 Faro, Portugal; 2CHANGE—Global Change and Sustainability Institute, Faculdade de Ciências e Tecnologia, Campus de Gambelas, Universidade do Algarve, 8005-139 Faro, Portugal; 3Instituto Nacional de Investigação Agrária e Veterinária, Quinta do Marquês, 2784-505 Oeiras, Portugal

**Keywords:** *Diplotaxis tenuifolia*, *Eruca sativa*, wild rocket, garden rocket, germplasm collections, SSR markers, SNP-CAPS markers

## Abstract

Rocket is the common designation for two baby-leaf salad crops of the Brassicaceae family: *Eruca sativa* (L.) Cav., usually referred to as annual garden rocket, and *Diplotaxis tenuifolia* (L.) DC. commonly named to as perennial wild rocket. *E. sativa* is used for human consumption since antiquity. However, the growing consumer preference for *D. tenuifolia* is being accompanied by the fast increase in its production area and commercialization of new cultivars. Nevertheless, the worldwide number of wild rocket accessions maintained in germplasm collections is very reduced, the solution for which situation the project “REMIRucula” intends to contribute, establishing a germplasm collection at the INIAV, Oeiras, Portugal. Herein, we report on the establishment via next generation sequencing (NGS) of the first genome assembly of *D. tenuifolia* and the identification of specific single sequence repeat (SSR) and single nucleotide polymorphisms (SNP) loci for the establishment of specific DNA-markers for this species. A representative set of 87 *D. tenuifolia* and 3 *E. sativa* accessions were assessed by 5 SSR and 9 SNP-CAPS markers, allowing a drastic discrimination between both species and the establishment of unequivocal molecular fingerprints for the analyzed accessions. The non-discrimination within six pairs and one trio of *D. tenuifolia* accessions is discussed.

## 1. Introduction

Rocket is the commonly used name to refer to two different species of baby-leaf salad crops, belonging to the Brassicaceae family: *Eruca sativa* (L.) Cav. (2*n* = 22) and *Diplotaxis tenuifolia* (L.) DC. (2*n* = 22).

*Eruca sativa*, also referred as cultivated or annual garden rocket, although used for human consumption for millennia, can be a problematic weed, as happens in China [1]. *D. tenuifolia*, also referred to as wild or perennial rocket, is an invasive weed in Europe, USA, Argentina, and is particularly noxious in Australia, where some states attempt to control its spread [2].

Nevertheless, both species are edible, tasty, and the search for new healthy food products by modern consumers has promoted their worldwide production and commercialization, particularly as ready-to-eat, pre-packaged salads.

The most evident distinctive morphologic traits that differentiate the two species are the white petals and simple silique of cultivated garden rocket vs. the wild rocket yellow petals and septate silique [3] (Figure 1).

Other differences, e.g., in seed size (much smaller in wild rocket), number of allowed successive cuttings, response to abiotic factors, and other aspects that determine some specificities of their production and commercialization were thoroughly reviewed by [4].

The two species produce glucosinolates, and multiple other secondary metabolites, namely antioxidants. A very inclusive comparative analysis of the data of the research carried out on both rocket species, regarding the variability and levels of these two kinds of compounds, and nitrate, crude fibers, total minerals, and carbohydrates, their variation depending on the environmental and storage conditions, and their effects on human health, was performed by [5]. A recent similar review, that includes the characterization of sensory properties and consumer preferences, was published by [6].

A wide range of information, specifically focused on the wild rocket (*D. tenuifolia*) biology, biochemical and nutraceutical properties, farming practices, crop protection, and industrial processing can be accessed in the review published by [7].

As previously noticed by [8], the available statistical data still do not consider the two rocket species separately. Nevertheless, the general perception is that the trend in the evolution of the relative production and commercialization of the rocket species is characterized by the increasing importance and share of *D. tenuifolia*, the production area of which attained 4800 ha in Italy [9].

As the importance of *D. tenuifolia* as a baby-leaf crop grows, plant breeding and germplasm conservation activities regarding this species become more imperative. Multiple companies are presently dedicated to breeding, producing, and commercializing new *D. tenuifolia* cultivars although, so far, the number of collected and conserved genotypes is still low. A brief consult of the International Minor Leafy Vegetables Database (https://ecpgr.cgn.wur.nl/LVintro/minorlv/con_spec.htm, accessed on 1 November 2022) shows that *D. tenuifolia* (plus *Diplotaxis* sp. and spp.) counts for 90 references vs. 663 registers for *E. sativa* (syn. *E. vesicaria*). Similarly, a consultation of the Genebank Information System of the IPK, Gatersleben, Germany (https://gbis.ipk-gatersleben.de/gbis2i, accessed on 1 November 2022) identifies 11 accessions of *D. tenuifolia* and 1 accession of *Diplotaxis* sp. vs. 145 accessions of *E. sativa*.

Among the main objectives of the collaborative project, REMIRucula is the establishment of a germplasm collection of *D. tenuifolia*, including some accessions of *E. sativa*, at the Instituto Nacional de Investigação Agrária e Veterinária (INIAV), Oeiras, Lisboa, Portugal. A representative number of accessions is intended to be tested for their response to a specific isolate of *Hyaloperonospora* sp. collected on a downy mildew naturally infected *D. tenuifolia* plant, and to be assessed for their genetic variability and genetic relationships by DNA markers and their unequivocal molecular identification by specific DNA markers.

The use of DNA markers in *Diplotaxis* studies has been relatively limited and performed using Inter Single Sequent Repeats (ISSR) and Random amplified Polymorphic DNA (RAPD) markers, mainly focused on the assessment of interspecific relationships [10] and identification of interspecific hybrids [11].

Recently, the genetic relationships among a large group of *D. tenuifolia* accessions of the novel germplasm collection, including some *E. sativa* accessions as the outgroup, were assessed using RAPD and ISSR markers [3]. These molecular analysis techniques have discriminated between the accessions of both species and allowed the genetic diversity within each species and the genetic relationships among the respective accessions to be assessed.

Herein, we describe the first wild rocket (*D. tenuifolia* (L.) DC.) genome assembly via next generation sequencing, the retrieving of some hundreds of single sequence repeat (SSR) and single sequence polymorphisms (SNP) loci, and the use of 14 of these markers for the unequivocal molecular characterization and identification of 90 accessions: 87 of *D. tenuifolia* and 3 of *E. sativa* accessions used as the outgroup.

## 2. Results

### 2.1. Next Generation Sequencing of Diplotaxis Tenuifolia

The next generation sequencing of the enriched with nuclear DNA sample of *D. tenuifolia* L. produced over 15.3 Gb which, using the Genomics Workbench v. 12.0.3, were assembled in 177,692 contigs totaling 424 M nucleotides (N50 = 4686; L50 = 23,088).

This first *D. tenuifolia* genome assembly was uploaded to the National Center for Biotechnology Information (NCBI) as “UALgDiploT.01” (www.ncbi.nlm.nih.gov/assembly/GCA_014822095.1)

### 2.2. SSR Markers Analysis

Five hundred SSR loci were identified among the new NGS-established genome assembly contigs and uploaded to the NCBI database (accessions: MT317952 to MT317453). Primer pairs were designed for 20 (Appendix A (Appendix A)) out of the 500 SSR loci and, after preliminary amplification tests (Figure 2), 5 out of the 20 SSR loci were selected for the unequivocal identification of 90 (87 wild rocket and 3 garden rocket) accessions of the rocket germplasm collection (Table 1).

The forward primer of each pair of the five SSR loci retained for further analysis was labelled with a fluorophore (Table 1) and, after a previous quality test in agarose gels, the amplified products were sent for fragment analysis by capillary polyacrylamide gel electrophoresis to the company STAB VIDA, Lisboa, Portugal.

The results of the fragment analysis were visualized graphically using the Peak Scanner™ Software v.1.0 (Thermo Fisher Scientific, Waltham, MA, USA) and the right alleles peaks were identified among the secondary peaks that result from the “band stuttering” [12], commonly associated with the SSR technique (Figure 3).

The fragment analysis allowed the identification of 70 different alleles for the 5 SSR loci that were permitted to perform the first assessment of the genetic similarities and genetic relationships between the analyzed accessions (Appendix A) and the establishment of specific SSR markers patterns (Appendix A) for identification of most of the *D. tenuifolia* accessions. Nevertheless, as can be observed in the respective dendrogram (Figure 4A), one cluster of eight accessions, three clusters of three accessions, and eight clusters of two accessions of this species shared identical SSR patterns, an issue that will be discussed below.

Although the primers were designed for the specific amplification of SSR loci identified in the *D. tenuifolia* genome, the *E. sativa* accessions amplified for three out of the five microsatellite markers, which discriminated among the three accessions of this species and contributed to their drastic discrimination from the wild rocket accessions.

The SSR marker MT317577 did not amplify in the three *E. sativa* accessions and in the six *D. tenuifolia* accessions: 56; 91; 92; 97; 134; and 146 (Appendix A). This is an interesting result, since, except for accessions 91 and 134, the other four accessions were previously found by RAPD and ISSR analyses to be relatively distanced genetically from the bulk of the accessions of the same species and closer, although still very apart, to the then analyzed *E. sativa* accessions [3].

### 2.3. SNP-CAPS Markers Analysis

The *D. tenuifolia* genome assembly “UALgDiploT.01” was additionally mined, using the software Geneious Prime v.2021.2.5, for the identification of sequences harboring single nucleotide polymorphisms (SNP). 

The genomic sequences, of approximately 500 nucleotides, surrounding the first identified 500 SNPs were retrieved and uploaded to the NCBI database (accessions: DiploTSNP.001 to DiploTSNP.500). The identified SNP loci were further analyzed using the software NEBcutter V2.0, New England Biolabs, Ipswich, MA, USA (http://nc2.neb.com/NEBcutter2/) for the identification of restriction enzymes that differently digested the alternative SNP alleles, allowing these markers to be assessed as CAPS (cleaved amplified polymorphic sequences) markers. Reflecting the method of analysis, the SNP markers are further referred to as SNP-CAPS.

Nineteen out of the selected 500 SNP loci were identified as harboring the polymorphic nucleotide within the sequence 5’-TCGA-3’ differentially recognized by the restriction enzyme TaqI, allowing this enzyme to be used for allele discrimination within these loci. Primers were designed for 9 SNP loci (Table 2 and Appendix A) which, after amplification and analysis by agarose gel electrophoresis (Figure 5), were shown to be suitable for molecular discrimination and were used to assess the 90 germplasm accessions.

The SNP loci, the specifically designed primers, the amplification annealing conditions, and the respective resulting fragments after TaqI digestion are displayed in Table 2. The results of the molecular analysis of the accessions by SNP-CAPS markers can be fully consulted in Appendix A.

The three *Eruca* accessions were amplified uniquely for the locus DiploTSNP.045 which, as for the large majority of the *Diplotaxis* accessions, was shown to be homozygous for the restriction by TaqI. Nevertheless, this circumstance did not hamper the *Eruca* accessions (29, 68, 105) to cluster apart from the *Diplotaxis* accessions (Figure 4B).

The analyzed SNP-CAPS markers were not enough for the full discrimination of the *D. tenuifolia* accessions, since only 24 accessions were characterized by individual patterns. The remaining 63 accessions remained gathered in clusters of full identity (DICE coefficient = 1.000) comprehending from 2 or 3 accessions to larger groups of 6 or 11 accessions (Appendix A; Figure 4B).

### 2.4. Combined Results of SSR and SNP-CAPS Markers

The combination of the results of the analyses performed with both types of markers allowed the analysis of the genetic similarities and genetic relationships between the 90 accessions to be refined (Appendix A; Figure 6).

Individual, specific molecular fingerprints were identified for almost all accessions, except for one trio and six pairs of wild rocket accessions that exhibited full similarity, an issue that will be discussed below. The immediately below high genetic-similarity (0.964) was exhibited by 17 pairs of accessions of this species.

The lowest genetic similarities reckoned between *D. tenuifolia* accessions were 0.464, between accession 58 and accessions 18 and 165, and 0.481 between accessions 56 and accessions 16, 18, and 165. However, the calculated genetic similarity values between two accessions of this species were over 0.6 in ~96% of the cases. As expected, the lowest coefficient of similarity (0.111) was registered between accessions of *Diplotaxis* and *Eruca* species.

The established molecular fingerprints (Appendix A) allowed clear discrimination between the two rocket species and among the accessions within each species, except for the below-discussed cases of identical molecular patterns of some *D. tenuifolia* accessions.

## 3. Discussion

In a previous assessment of the genetic relationships among a large set of accessions of the same germplasm collection using RAPD and ISSR markers, only in one case did the genetic similarity (DICE coefficient) between two *Diplotaxis* accessions fell below 0.7, and in 95% of the cases, this parameter was over 0.8 [3].

This level of genetic similarity agrees with the results obtained in our lab during the last few decades whenever randomly amplified DNA markers (RAPD, ISSR, AFLP) were used to assess the genetic similarity and genetic relationships within multiple plant species, e.g., pear [13]; apple [14]; fig [15]; or common beans [16].

In the present study, as expected, the calculated genetic similarity values based on SSR markers were relatively low. This is a consequence of the hyper-polymorphism of these markers, that makes them highly useful for the unequivocal identification and discrimination of individuals, and for the establishment or categorical abolishment of close genetic relationships between individuals (e.g., paternity tests), but not suitable for the quantification of these relationships. This last circumstance is clear in the present study (Appendix A) as the reckoned genetic similarity between accessions of the same species often fell drastically below 0.8, reaching values as low as 0.3 (e.g., accessions 12 and 16 of *D. tenuifolia*).

On the other hand, the intraspecific (*D. tenuifolia*) genetic similarity values calculated based on the SNP-CAPS markers were higher, with the lowest value (0.556) registered in eight cases (Appendix A).

In comparison with the results obtained in our lab using randomly amplified markers [3], the low discriminating results of SNP-CAPS markers analyses are, apparently, a consequence of the low number of analyzed loci. This situation could be improved by the assessment of a larger number of SNP loci or by complementation with results of other molecular markers analyses. We chose the second option and the combination of the results of the SNP-CAPS and SSR markers analyses allowed the identification of specific molecular fingerprints for most of the accessions (Appendix A; Figure 6).

The seven cases of non-discrimination among *D. tenuifolia* accessions deserve a detailed analysis and discussion, as the identification of specific fingerprints for all accessions of the germplasm collection is one of the main objectives of this work.

Accession 1 was provided in 2015 identified as a commercial variety by a wild rocket producer, while accession 98 was provided in 2019 as the same cultivar by the breeding company.

Accessions 160 and 161 appeared as identical in our analysis However, they were supposed to correspond to two different cultivars provided by the same rocket producer. For that reason, an accurate comparative analysis of the multiple phenotypic traits of these two accessions will be performed, new samples will be requested from the original providers, and a new molecular analysis will be carried out. Then, the needed correction will be introduced in the germplasm collection data.

Accessions 17 and 24 were provided in different years by the same donor, under the same name. These accessions were assumed as an internal control of the performed analyses.

Accessions 10 and 19 were provided, respectively, by a wild rocket producer and a breeding company as the same, relatively resistant to downy mildew cultivar.

Accessions 18 and 165 were provided by two different wild rocket producers, one in Portugal and the other in the USA, as the same downy mildew-resistant cultivar.

Accession 39 was registered (in 2019) in the germplasm collection as a trial sample of a new cultivar provided by a breeding company. Accession 164 was received later (in 2020) and identified as a commercially available downy mildew-resistant cultivar. Both accessions are among the very few that, in our previous studies [3], have exhibited strong resistance to the used *Hyaloperonospora* isolate.

Accessions 23, 38, and 158 were provided by three different donors. The first from a wild rocket leaf producer. The second as a new cultivar under assay (identified by a company code). The third accession as a very well identified, highly resistant to downy mildew, cultivar. In fact, the three accessions were also among the very few *D. tenuifolia* accessions that exhibited downy mildew resistance in our previous studies [3]. This relatively rare phenotypic trait shared by the three accessions, also contributed to confirm these three accessions as the same cultivar, despite the different providers.

Except for the case of the molecular identity of the accessions 160 and 161, which needs further explanation, the other cases of accessions that exhibit the same molecular pattern reinforces our assumption that the set of molecular markers used in this study can be further utilized for the unequivocal identification and register of all accessions of the germplasm collection.

## 4. Materials and Methods

### 4.1. Plant Germplasm Accessions

A set of 87 *D. tenuifolia* and 3 *E. sativa* accessions of the germplasm collection (INIAV), previously tested for their interaction with the isolate D5 of *Hyaloperonospora* sp., the causing agent of downy mildew disease [3], were selected for unequivocal molecular characterization and identification by specific SSR and SNPs patterns.

### 4.2. Plant Growing Conditions

The seeds were quickly washed with tap water and common detergent and immersed for 1 min into a disinfection solution containing 0.5% of SDS and 10% of commonly commercialized bleach. After thorough washing with distilled water until the total removal of the disinfection solution, the seeds were transferred to Petri dishes containing three layers of filter paper saturated with tap water. During the following days, the consecutively germinated seedlings were transferred in groups of 3 to small, 9 cm diameter pots containing a mix of 50% peat and 50% perlite. Five pots per accession were transferred to a glass greenhouse, and 3 weeks later only one well-succeeded plant was left to grow per pot.

### 4.3. DNA Extraction

#### 4.3.1. DNA Extraction for Molecular Characterization

The genomic DNA extraction for molecular markers analysis was performed as described in [17] with minor modifications. One leaf from 5 plants of each accession was removed, washed with tap water, and wiped up with paper. The central nervures were removed using a scalpel and the leaves were grounded under liquid nitrogen in a mortar with a pestle. The obtained fine powder was transferred to a microfuge tube containing 500 µL of extraction buffer (250 mM Tris-HCL, pH 8.0, 25 mM EDTA, 1% SDS) until the final volume of the suspension reached approximately 750 µL. RNase A (20 μg.mL^−1^) was added to the tube and then transferred to a water bath at 65 °C for 15 min. Already at room temperature, 1 volume of phenol: chloroform: isoamyl alcohol (25:24:1) was added to the tube which, after successive inversions for 1 min, was centrifuged at 13,000 rpm for 3 min. The upper phase was transferred to a new tube and extracted with 1 volume of chloroform: isoamyl alcohol (24:1) as described in the previous step. This second extraction was repeated at least once, until the interphase appeared completely transparent. The final upper phase was transferred to another tube and mixed and precipitated with 3 volumes of cold absolute ethanol and kept at −20 °C. After centrifugation at 13,000 rpm, the DNA pellet was dried and slowly resuspended for two days in 50 µL of TE0.1 (10 mM Tris, 0.1 mM EDTA) in a refrigerator.

#### 4.3.2. DNA Extraction for NGS Sequencing

All described procedures were carried out under cold conditions. Five leaves from accession 7 were excised, prepared, and ground as described in the previous paragraph. The obtained fine powder was resuspended in 25 mL of nuclei isolation buffer (50 mM Tris—HCl pH 8.0, 1 M saccharose, 25 mM MgCl2, 100 mM KCl, and 2% Triton X-100). After very mild agitation for 1 min the suspension was filtered through an inox sieve (mesh size 75 µm) helped by the addition of 10 mL of isolation buffer. The filtrate was divided equally into two 15 mL sterilized Falcon tubes and centrifuged at 30 g in a bench centrifuge for 5 min. The supernatants were collected to two new Falcon tubes and centrifuged again at 1100 g for 5 min. The supernatants were discarded, and a small amount of the nuclei-enriched pellets quickly collected using a micropipette tip and mixed with 10 µL of a DAPI solution on a glass microscope slide for quality analysis under UV microscopy (Olympus Vanox AHBT3). The pellets were resuspended in 1 mL of DNA extraction buffer and the DNA was extracted as described above.

### 4.4. Quantification and Quality Evaluation of the Extracted DNA

The integrity of the extracted DNA and eventual contamination with RNA were assessed by agarose gel (1.4%) electrophoresis. The DNA concentration was determined approximately in the same gels by comparison with different known amounts of genomic DNA extracted from *Pisum sativum* roots, which do not contain chlorophyll or other pigments usually present in leaf samples that can bias the spectrophotometry results. A more accurate quantification was then obtained by UV spectrophotometry (NanoDrop One; Thermofisher, Waltham, MA, USA). The respective values, which can be biased by the fluorescence of the remaining leaf pigments (chlorophyll) and the presence of other contaminants, were accepted if falling within the concentration limits established in agarose gels. The amplifiability of the DNA samples was assessed by RAPD-PCR, performed as previously described in [3].

### 4.5. NGS Sequencing

The precipitated with three volumes of absolute ethanol genomic DNA was centrifuged, washed with 75% ethanol, centrifuged again and the pellet was left to dry for two hours at room temperature. After slow resuspension in autoclaved milli-Q water, the DNA integrity and purity were assessed as above described. After quantification using UV spectrophotometry (NanoDrop One), the DNA was sent to STAB VIDA Lda, the company requested for next generation sequencing using an Illumina HiSeq platform.

### 4.6. Primer Design and Synthesis

All primers were designed, their parameters calculated and their eventual self- or pair-annealing assessed, using the FastPCR 6.7 Software [18]. Common, non-labeled, primers were synthesized by the company Eurofins Genomics (Ebersberg, Germany). The fluorescent labeled primers were ordered from the company to STAB VIDA (Lisboa, Portugal).

### 4.7. Single Sequence Repeats (SSR) Markers Analysis

The SSR loci (~500 bp sequences containing an SSR motif) were identified manually by a random search for microsatellite motifs among the sequence contigs. The fragment analysis of the amplified by fluorescent primers SSR markers was performed in a 3730XL Genetic Analyzer platform using GeneScan™ 500 LIZ™ (Thermofisher, Waltham, MA, USA) as the dye size standard. The resulting data were analyzed using the Peak Scanner™ Software v. 1.0 (Applied Biosystems, Thermofisher, Waltham, MA, USA).

The amplification of the SSR (microsatellite) markers was performed in 30 µL reactions, starting with an initial denaturation at 94 °C for 1 min and 30 sec, followed by 35 cycles of 30 sec denaturation at 94 °C; 30 sec annealing, at different temperatures depending on the specific primer pair, and 1 min extension at 72 °C, followed by a period of final extension at 72 °C for 10 min.

The PCR products were analyzed in 3% agarose gel electrophoresis and the better amplified markers were selected for further, more accurate, analysis among the studied accessions. The amplifications were repeated with the same pair of primers with the forward primer labeled with a fluorochrome. Half of the amount of the amplified products was analyzed by agarose gel electrophoresis and the second half of the approved amplified samples was sent to the company STAB VIDA for fragment analysis by capillary polyacrylamide gel electrophoresis.

### 4.8. Single Nucleotide Polymorphisms (SNP) Markers Analysis

Five hundred SNP loci were identified using the Geneious Prime v.2021.2.5 software (Dotmatics, Boston, MA, USA). Seven nucleotide sequences, three nucleotides from each side of the identified SNP, were analyzed by the NEBcutter V2.0 software [19] for the identification of restriction enzymes that differentially recognized the alternative SNP alleles.

Nineteen SNP markers harboring a TaqI restriction site encompassing the polymorphic nucleotide were selected for further work and amplified using the same protocol used for the SSR markers. Fifteen microliters of the amplified products were analyzed by 3% agarose gel electrophoresis. The remaining 15 µL of well amplified samples were then cut with the TaqI restriction enzyme, and the samples were analyzed as CAPS (cleaved amplified polymorphic sequences) markers in 3% agarose gels.

### 4.9. Data Analysis

The detailed analysis of the sequence contigs and respective reads were performed using the software Tablet 1.21.02.08 (The James Hutton Institute, Aberdeen, Scotland, UK) [20].

The NTSYS-pc program [21] was used for cluster analysis. The genetic similarity between the accessions was reckoned using the coefficient DICE [22] by pairwise comparisons based on the percentage of common fragments, according to the following equation: similarity = 2Nab/(Na + Nb), where Nab is the number of scored amplification products simultaneously present in accessions ‘a’ and ‘b’, Na is the number of amplification products scored in accession ‘a’, and Nb is the number of scored fragments in accession ‘b’. The unweighted pair-group method with arithmetic averages (UPGMA) was used to calculate the cophenetic matrix used for dendrogram construction. The cophenetic correlation coefficient (r) was calculated by comparison of the similarity matrix with the UPGMA-produced cophenetic matrix graphically represented as a dendrogram.

## 5. Conclusions

The first genome assembly for wild rocket (*Diplotaxis tenuifolia* (L.) DC) here described and the provided information regarding multiple SSR (microsatellite) and SNP loci constitute major research tools available for the scientific community engaged in genetic and genomics studies on rocket species.

The combined use of genome-specific SSR and SNP-CAPS allowed the identification of specific molecular patterns for almost all analyzed *D. tenuifolia* accessions, the identification and confirmation of cases of synonymy, and the clear discrimination from the *E. sativa* accessions.

The SSR and SNP-CAPS markers, tested and validated in this study, will be used for the unequivocal identification of all present and upcoming accessions of the Portuguese germplasm collection of *D. tenuifolia*, and for the initial research steps towards the genome location of downy mildew-resistance genes in this species.

## Figures and Tables

**Figure 1 plants-11-03482-f001:**
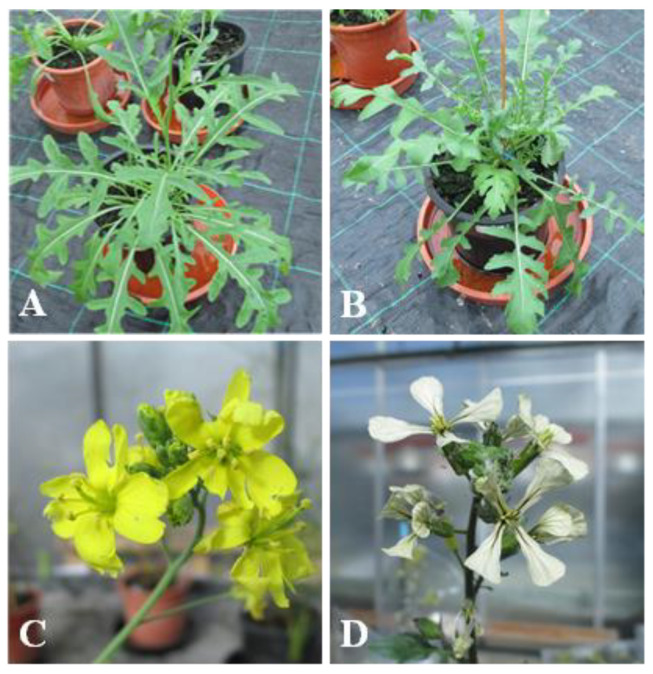
(**A**) *Diplotaxis tenuifolia*; (**B**) *Eruca sativa*; (**C** and **D**) Respective inflorescences.

**Figure 2 plants-11-03482-f002:**
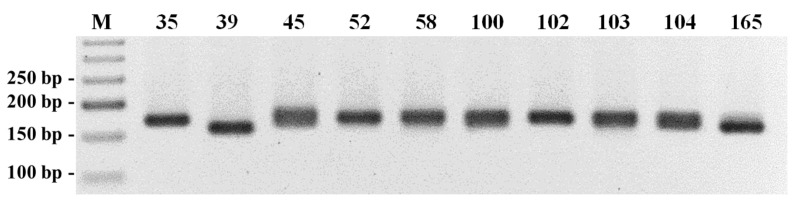
Agarose (3%) gel electrophoresis of the preliminary amplification of the SSR locus MT317527 among 10 *D. tenuifolia* accessions.

**Figure 3 plants-11-03482-f003:**
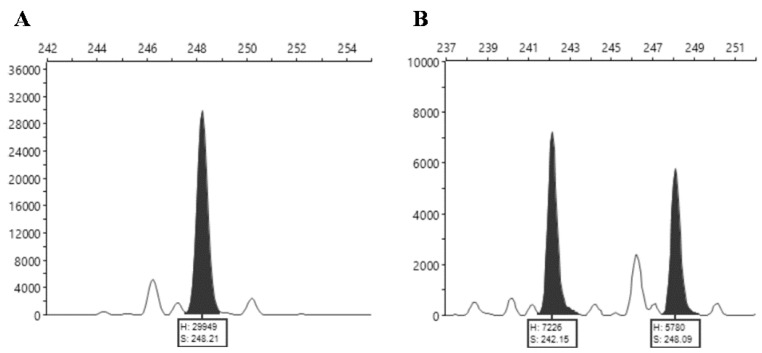
Capillary polyacrylamide gel electropherograms of the locus MT317577 (**A**) Homozygous pattern of accession 15; (**B**) Heterozygous pattern of accession 37. X axis—Fragment size (bp); Y axis—Relative Fluorescence Unit (RFU). Notice the associated smaller “stutter” peaks.

**Figure 4 plants-11-03482-f004:**
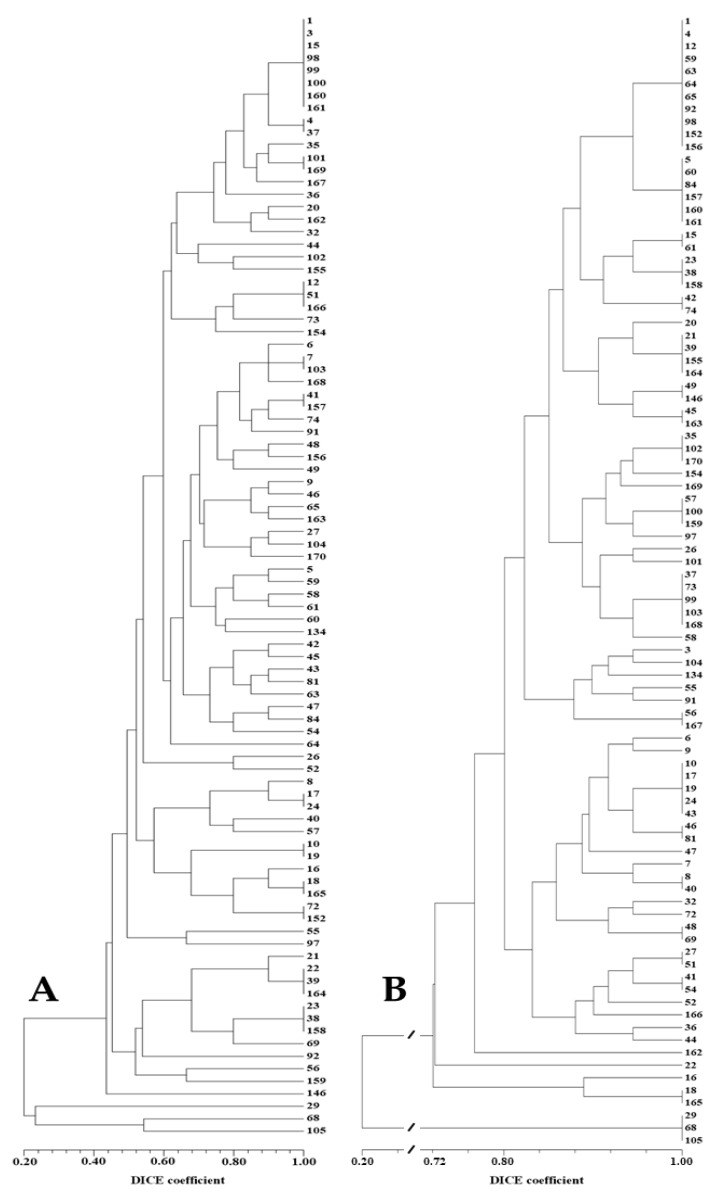
(**A**) Dendrogram depicting the genetic relationships among the 90 germplasm accessions established based on SSR markers. The *E. sativa*. accessions (Nos. 29, 68, and 105 at the bottom) are clustered apart from all other (*D. tenuifolia*) accessions. (**B**) Dendrogram displaying the genetic relationships among the 90 germplasm accessions established based on SNP-CAPS markers.

**Figure 5 plants-11-03482-f005:**
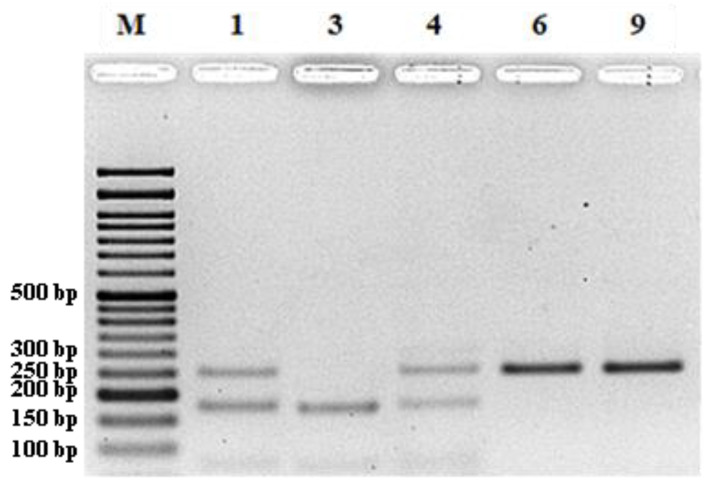
Agarose (3%) gel electrophoresis analysis of the molecular patterns of 5 accessions for the DiploTSNP.290 locus. Accessions 1 and 4 (Heterozygous Y/N); Accession 3 (Homozygous Y/Y); Accessions 6 and 9 (Homozygous N/N); M—DNA ladder VI (Nzytech). Y—one allele restricted; N—one allele not restricted.

**Figure 6 plants-11-03482-f006:**
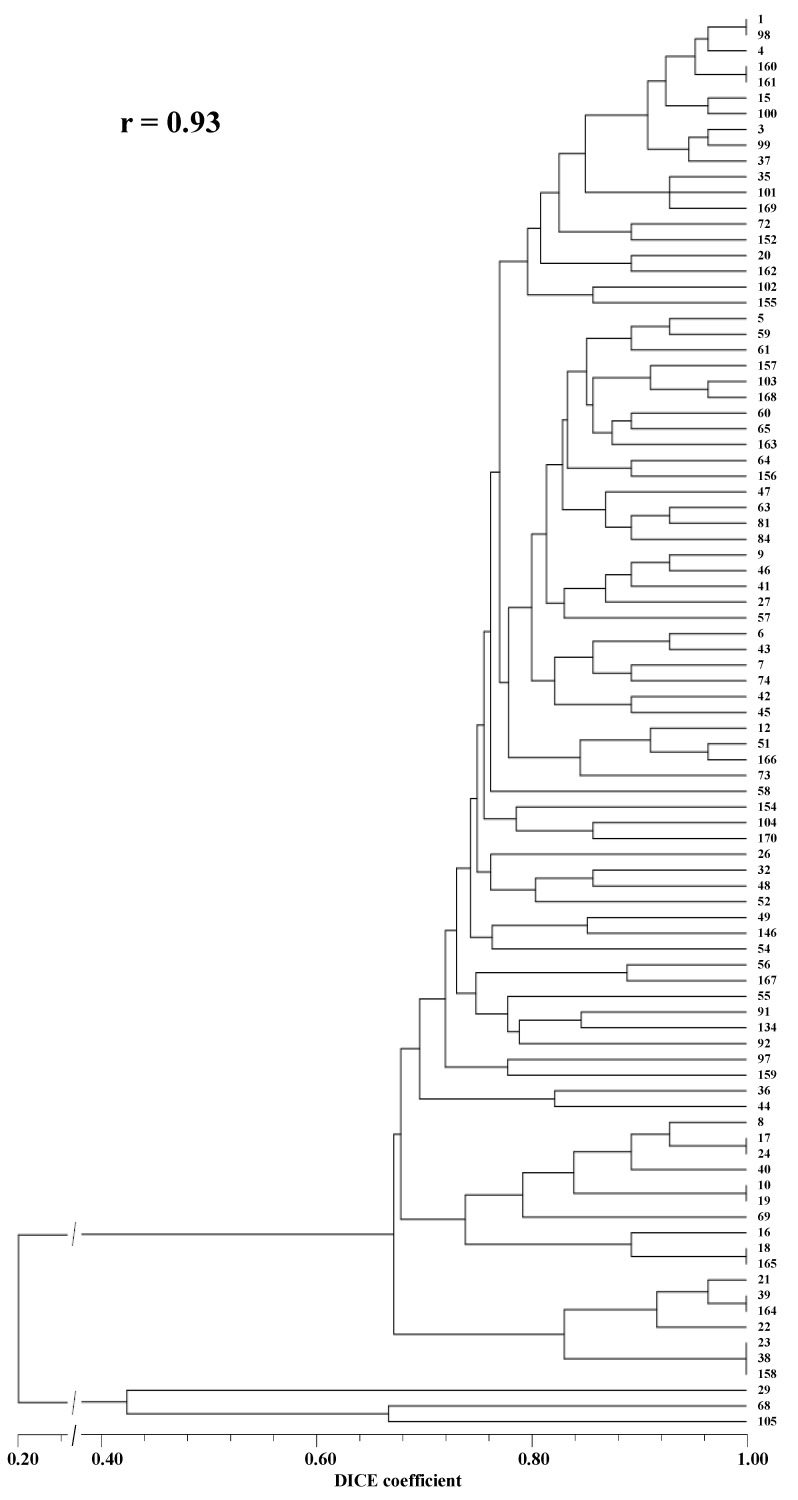
Dendrogram depicting the genetic relationships among the 90 germplasm accessions established based on SSR and SNP-CAPS markers. The *Eruca* sp. accessions (29, 68, and 105 at the bottom), used as an outgroup, are clustered apart from all the other (*D. tenuifolia*) accessions.

**Table 1 plants-11-03482-t001:** SSR Loci Used for Accessions Identification.

Locus	Primers		T (°C) **	Fluorophore
MT317577	FW	CGGATAAACATATCCGCTT *	57	Atto 565
	RV	GGTTAACATCACTAACGGT		
MT317610	FW	GTCTATTGATCTGATGCCG *	57	Atto 550
	RV	GCGAGAACCGATCTATGA		
MT317823	FW	CCAACATAGAAAGGTGCG *	57	Atto 550
	RV	GAGTTCCTCCAAAAGCTG		
MT317527	FW	ACTTTGACGAAACGAAGC *	58	HEX
	RV	CTCAGAACCAAAGAGAAGC		
MT317537	FW	CAGCTTTCTGTTAGTGGTC *	59	HEX
	RV	CCAACCAAAAACGTCAGA		

* Fluorophore labelled; ** Annealing temperature.

**Table 2 plants-11-03482-t002:** SNP-CAPS Markers Used for Accession Identification.

Locus	Primers		T (°C) *	SNP	Restriction Fragments
DiploTSNP.045	FW	GTCCCATGATTAGATATGGT	58	G/T	143/80
	RV	AGTCTTTAAGTGACAAGCG			
DiploTSNP.062	FW	GCGTTTGAATGGTTTCTG	56	T/C	205/75
	RV	CTTTGCTCATCCGTCAA			
DiploTSNP.090	FW	CATCTGTGCCTGATCTC	58	G/T	191/54
	RV	CATGTCAAGTCGTGATTCA			
DiploTSNP.138	FW	GAGCCTTGTAACAGATCC	59	T/G	194/94
	RV	CTGGGTTTGTGACTGTTG			
DiploTSNP.213	FW	TGGCATCCTACTCTTTCTC	60	C/T	170/61
	RV	CCGTGTGTTCATGATATGG			
DiploTSNP.220	FW	TCTACGATGCCAAGAGTT	58	T/C	144/94
	RV	TGTACTGGAAAGGTATCCC			
DiploTSNP.270	FW	CTAATGAGTTCGTCGTTCG	60	A/G	216/108
	RV	AGCTTATCTCTTCCTGGTG			
DiploTSNP.290	FW	CGAAAGAGATGCTATCGG	59	C/T	167/76
	RV	TACGTCCTCAGTTCTAACC			
DiploTSNP.448	FW	GGCTGGATATGTGTCAGA	59	A/C	155/127
	RV	GTCATGTTCCTTCTGGAAC			

* Annealing temperature.

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
