# Peer review of "Assessment of Wild Rocket (Diplotaxis tenuifolia (L.) DC.) Germplasm Accessions by NGS Identified SSR and SNP Markers"

_plants, 2022, doi:10.3390/plants11243482_

Round 1

Reviewer 1 Report

I think the manuscript shows good information that is useful for more studies and the improvement of the baby-leaf salad crops studied here. The major problem I detected relates to mistakes of misspellings, many double spaces, and grammar. I highly recommend correcting those that I pointed out in the manuscript as "track changes"

Author Response

Dear Reviewer1 

Thank you very much for your comments, detailed analysis, corrections  and recommendations regarding the manuscript.

We have accepted all your suggestions and made additional minor corrections to the manuscript.

Thank you again.

José Leitão

Reviewer 2 Report

In this report, the authors used next generation sequencing to identify specific single sequence repeat (SSR) and single nucleotide polymorphisms (SNP) for the establishment of specific DNA-markers for Diplotaxis tenuifolia. Authors have smartly combined SNP-CAPS and SSR markers to identify specific molecular fingerprints for most of the accessions because the result from SNP-CAPS alone provides low number of analyzed loci. With the help of 5 SSR and 9 SNP-CAPS they access 87 accessions of D. tenuifolia and 3 accessions of E. sativa. However, authors should address some points that will help readers to understand their work properly.

Did the authors find any phenotypic characters associated with the markers for the accessions? If so, provide details of that in the manuscript so that it can be used for QTL analysis.  

The number of references are less in the introduction and discussion section which can be increased

The sentence in lines 69-70 is not clear, please consider rewriting this sentence.

Remove the word “will” from line number 72

Write Genus and Species names uniformly in italics, in the whole manuscript.

Close the bracket of “apple [14” in line 213. 

Label A and B in Figure 4.

Author Response

Dear Reviewer 2

Thank you very much for your comments, corrections  and recommendations regarding the manuscript.

Below, please find the response to your  questions and  suggestions.

1) Did the authors find any phenotypic characters associated with the markers for the accessions? If so, provide details of that in the manuscript so that it can be used for QTL analysis.  

Answer: Not so far. The relationship between the main phenotypic trait of interest for our studies recently started. F1 (pseudo testcross) and F2 progenies will be used for that purpose.

2) The number of references are less in the introduction and discussion section which can be increased.

Answer: We understand that the number of citations seems low. However, most of the works published on D. tenuifolia (and on E. sativa) are mainly focused on metabolite contents which is not the topic of this work. Nevertheless, references to comprehensive reviews on these topics are provided to readers that can access the large information contained in the referred to articles.

3) The sentence in lines 69-70 is not clear, please consider rewriting this sentence. Answer: Done. Thank you.

Remove the word “will” from line number 72. Answer: Done. Thank you.

Write Genus and Species names uniformly in italics, in the whole manuscript. Answer: Done. Thank you.

Close the bracket of “apple [14” in line 213.  Answer: Done. Thank you.

Label A and B in Figure 4. Answer: Done. Thank you.

Thank you very much again for your criticisms and suggestions

José Leitão